# Brain Response to Interferential Current Compared with Alternating Current Stimulation

**DOI:** 10.3390/brainsci13091317

**Published:** 2023-09-13

**Authors:** Zonghao Xin, Yoshifumi Abe, Akihiro Kuwahata, Kenji F. Tanaka, Masaki Sekino

**Affiliations:** 1Department of Bioengineering, School of Engineering, The University of Tokyo, Tokyo 113-8656, Japan; zonghao.xin@riken.jp; 2Division of Brain Sciences, Institute for Advanced Medical Research, Keio University School of Medicine, Tokyo 160-8582, Japan; abeyoshihumi@gmail.com (Y.A.); kftanaka@keio.jp (K.F.T.); 3Department of Electrical Engineering, Graduate School of Engineering, Tohoku University, Sendai 980-8579, Japan; akihiro.kuwahata.b1@tohoku.ac.jp

**Keywords:** interferential current stimulation, alternating current stimulation, temporal interference, fMRI

## Abstract

Temporal interference (TI) stimulation, which utilizes multiple external electric fields with amplitude modulation for neural modulation, has emerged as a potential noninvasive brain stimulation methodology. However, the clinical application of TI stimulation is inhibited by its uncertain fundamental mechanisms, and research has previously been restricted to numerical simulations and immunohistology without considering the acute in vivo response of the neural circuit. To address the characterization and understanding of the mechanisms underlying the approach, we investigated instantaneous brainwide activation patterns in response to invasive interferential current (IFC) stimulation compared with low-frequency alternative current stimulation (ACS). Results demonstrated that IFC stimulation is capable of inducing regional neural responses and modulating brain networks; however, the activation threshold for significantly recruiting a neural response using IFC was higher (at least twofold) than stimulation via alternating current, and the spatial distribution of the activation signal was restricted. A distinct blood oxygenation level-dependent (BOLD) response pattern was observed, which could be accounted for by the activation of distinct types of cells, such as inhibitory cells, by IFC. These results suggest that IFC stimulation might not be as efficient as conventional brain modulation methods, especially when considering TI stimulation as a potential alternative for stimulating subcortical brain areas. Therefore, we argue that a future transcranial application of TI on human subjects should take these implications into account and consider other stimulation effects using this technique.

## 1. Introduction

Temporal interference (TI) stimulation is a potential noninvasive method that promises stimulation in encephalic regions by delivering multiple electric fields at relatively high frequencies but with a slight frequency offset [1]. Considering the resistive and capacitive properties of a lipid bilayer membrane, neural membranes generally possess intrinsic low-pass filtering of electrical signals [2], and therefore, neural electrical activity would be unable to follow very high-frequency oscillating electric fields. Thus, during TI stimulation, the neurons of the brain do not react directly to these high-frequency electric fields. However, owing to the small frequency offset (e.g., 10 Hz), the two applied fields would consequently merge into a temporally interfering field with amplitude modulation oscillating at different frequencies that the neurons can follow.

To date, the accumulated evidence has demonstrated the feasibility of this nascent brain stimulation concept both theoretically and empirically; the electric field distribution in the brain tissue produced by the stimulus current was investigated, demonstrating that the induced field intensity in the mouse brain could reach a value sufficient to cause an acute neuronal response [3]. Moreover, this performance of TI stimulation could be enhanced by exploiting a multielectrode strategy [3] and by applying rigid numerical criteria to optimize the produced electric field in the regions of the target [4,5]. An analogous optimization process was initially proved to be feasible in conventional transcranial electrical stimulation methods [6], but the optimal conditions were inconsistent across subjects [7].

On the other hand, animal experiments have been conducted exclusively in a few studies. The expression of c-fos in cortical and hippocampal structures was assessed to reveal the modulated brain regions immediately after performing TI stimulation [1]. A similar analysis benchmark was leveraged by Song et al., aiming to identify the activation of motor cortical neurons in combination with measurements of the myoelectricity signal in the forepaw [8]. In addition, TI stimulation was demonstrated to evoke seizurelike events in the CA field of the hippocampus in mice [9]. However, these pieces of evidence of the effectiveness of TI stimulation have been obtained mostly through numerical simulations and immunohistology, without considering the acute in vivo response of the neural circuit. Consequently, brainwide activation patterns in response to TI stimulation remain undefined, and the underlying mechanisms of the method have not been explicitly addressed.

We therefore focused on the fundamental premise of TI, where an interferential current (IFC) is assumed to be exerted on the neurons and consequently modulate the brain network through suprathreshold activation [1]. The recruitment of a neural cell by IFC has been validated using numerical simulation; a single neuron computational model based on the Hodgkin–Huxley theory has been applied to investigate the neurobiological mechanisms upon TI stimulation [10], showing that IFC may not recruit all neuron types, and the mechanisms of TI stimulation are more likely to be based on a network rather than single neuron dynamics. A similar study was performed based on a compartmental model of a myelinated mammalian axon to investigate the axonal mechanisms of TI stimulation [11]. The results showed that the activation of neurons by IFC could not be explained entirely by passive membrane filtering characteristics, as it was hypothesized previously [1], because the amplitude-modulated waveforms of IFC do not contain any low-frequency components. We therefore initially aimed to address a fundamental issue, that is, whether stimulation using IFC (the essence of TI stimulation) could induce neural modulation effects (especially suprathreshold activation) inside brain networks.

Another intriguing aspect of TI is the similarity of the method with transcranial alternating current stimulation (tACS), where an alternating current (AC) is generally applied above the scalp to entrain the neural activities. tACS is generally considered to be able to engage and interact with endogenous neural oscillations [12,13]. Comparative research between TI stimulation and tACS using a computational cortex model suggested that the oscillation of cortical cells could be synchronized to the envelope of the subthreshold TI stimulation waveform analogous to a conventional low-frequency tACS modulation [14]. However, compared with conventional tACS, substantially higher current intensities were required for TI to induce a phase synchronization of similar intensity. Moreover, these modulation effects could possibly be produced in deep brain regions [15] but would be contingent on the neuronal membrane temporal aspects. Therefore, it is also noteworthy to interrogate whether IFC could replicate the whole brainwide modulation effects as induced by conventional stimulation protocols or not. Here, we aimed to compare the brain activity in response to IFC stimulation and invasive low-frequency alternating current, where we evaluated the activation threshold and activated/inhibited brain regions upon both stimulation protocols on the rat models’ primary motor cortex (M1), using a functional imaging (fMRI) technique. To avoid ambiguity, the protocols applied in this study are termed as interferential current stimulation (IFC stimulation) and alternating current stimulation (ACS), both of which were invasively (which is distinct to the TI and tACS) implemented through implanted electrodes.

## 2. Materials and Methods

### 2.1. Animals

Experiments were performed using 16 adult male Wistar rats (10 weeks) weighing 270 ± 20 g. The sample size was determined based on our previous work [16,17] and “power analysis” using the “G*Power” software [18] (Version 3.1; effect size = 0.8, power value of 0.95, and α = 0.05). Before the experiment, animals were maintained in individually ventilated cages and housed in groups of two rats per cage. Food and water were provided *ad libitum* in a temperature- and humidity-controlled facility on a 12 h light/dark cycle. Two rats were used in the preliminary experiment to test the intensity of stimulation exerted on the motor cortex and to obtain structural images via MRI. The remaining rats (*n* = 14) underwent craniotomy and were used in fMRI experiments to inspect the stimulation-evoked cortical blood-oxygenation-level-dependent (BOLD) signals. Animals were immediately sacrificed after the two stimulation sessions.

### 2.2. Stereotaxic Surgery

During surgery, the animals were initially anesthetized with isoflurane (2%) mixed with air and then injected with urethane (Sigma-Aldrich, Saint Louis, MO, USA) with a dose of 1.25 g/Kg [19]. After complete anesthesia, the surgical field on the scalp was shaved and sterilized with 70% ethanol. The rats were then placed in a stereotaxic frame to implant the stimulator electrode. The temperature of the animal was maintained at 37 °C (monitored by a rectal thermostat probe) using a heating pad (Natsume Seisakusho Co., Tokyo, Japan). The scalp was incised to expose the cranium, and a small hole was drilled through the skull at the position above the right M1 (2.0 mm mediolateral, 1.0 mm anteroposterior from the bregma) [20]. Next, a glass-made tube (Kenneth, Tokyo, Japan) with a 2.1 mm outer diameter and a 1.4 mm inner diameter was affixed to the skull using Aron Alpha A adhesive (Sankyo Co., Ltd., Tokyo, Japan), following which a silver wire (Sigma-Aldrich, USA) of 0.25 mm diameter was inserted into the cortical tissue with a depth of 2 mm from the surface of the skull [1] (as demonstrated by Figure 1A).

After the surgery, the rats were transferred and mounted on the MRI bed fixed with ear bars for fMRI data acquisition. The animals’ body temperature was maintained at 37 °C throughout the experiments (CIRCULATING THERMO, Bruker Optik, Ettlingen, Germany). Body temperature was metered by an ECG/TEMP module (SA Instruments Inc., Stony Brook, NY, USA), and respiration was monitored by a respiration module (SA Instruments Inc., USA). Once the fixation was completed, a saline electrode gel was used to attach a cloth-based electrode with a 15 mm diameter conductive area (Bs-150, Nihon Kohden Corporation, Tokyo, Japan) to the dorsal skin of the animal (SignaGel, Parker Laboratories, Fairfield, NJ, USA).

### 2.3. fMRI Experiments

The fMRI data acquisition in this study was in accordance with the protocol we described previously [16] except for the following modifications: anatomical MRI (no stimulation) was acquired with the following parameters: rapid acquisition with relaxation enhancement (RARE); TR/TE = 2500/33 ms, FOV = 24 × 24 mm^2^, matrix = 256 × 256, slice thickness = 0.8 mm, slice gap = 0.00 mm, slice number = 14, repetition time = 6, scan time = 6 min. fMRI data were acquired with the following parameters: gradient echo (GE)-EPI sequence in the same position as with the anatomical MRI sequence; repetition time (TR)/echo time (TE) = 2000/15 ms, segments = 1, FOV = 24 × 24 mm^2^, matrix = 60 × 60, slice thickness = 0.8 mm, slice gap = 0.00 mm, slice number = 14, repetition number = 205, scan time = 6 min and 50 s. Before the actual image-taking procedure, a test image was acquired using the same MRI sequence of the structural image-taking process but without repetition. The slice number was reduced to four, allowing the scan to focus exclusively on the area where the electrode was implanted to confirm that the electrode was positioned precisely in the target region.

The stimulation of M1 was devised as a block design with interleaved trials of stimulation off (10 s) and stimulation on (60 s), repeated five times for a total duration of 6 min and 50 s (Figure 1C). The electric current was designed as either an interferential wave by summarizing a 2 kHz and 2.01 kHz sinusoidal wave (with an amplitude modulated envelope of 10 Hz) or a simple alternating current wave of 10 Hz. The stimulus was generated by a four-channel stimulus generator (STG-400, Multi Channel Systems MCS GmbH, Kusterdingen, Germany) and in-house software written in MATLAB 2022 (MathWorks, Natick, MA, USA). MRI data acquisition was synchronized with this stimulation protocol using a custom trigger device.

All the animals underwent both stimulation protocols with a random order (either IFC first or ACS stimulation first); fMRI data of respective rats were collected succeedingly for both stimulation protocols. Based on the preliminary trials, the stimulation current intensity was determined as an ascending order for ACS fMRI data to comprise three sessions with a current intensity of 0.3, 0.4, and 0.5 mA. As for IFC stimulation, the stimulus was applied with the intensity increasing from 0.4 to 1.0 mA with an interval of 0.2 mA.

### 2.4. fMRI Data Processing

Preprocessing and statistics were performed using SPM12 and an in-house software written in MATLAB 2022 [16]. The initial image preprocessing was performed individually for each animal. The RARE reference images were coregistered and spatially normalized to a standardized structural brain template for rats [21]. The time series images were realigned to correct for residual head motion, corrected for slice timing, and coregistered to the reference structural images. Subsequently, all images were spatially normalized and coregistered to the standardized brain structure template using the normalized parameters of the structural image. Finally, these images were resliced to a resolution of 0.4 × 0.4 × 0.8 mm^3^ and smoothed with a full width at a half maximum Gaussian kernel of 1.0 mm. After the preprocessing procedure, a voxel-based general linear model (GLM)–based statistical parametric analysis was individually employed to depict the activation maps.

Region of interests (ROI) analysis was conducted based on the GLM fitting results, where activation maps were overlapped with a delicately segmented atlas [22] to identify the specific brain regions excited by the stimulus. Exclusively, signal variation of the brain voxels with activation was extracted to depict the BOLD time course. Time series BOLD activation maps were individually calculated with respect to the baseline averaged over the 20 s period prior to stimulation and were averaged over the 5 stimulation periods. The BOLD peak intensities, defined as the maximum signal change with respect to baseline, were identified and averaged over the 5 stimulation periods. The time to peak was defined as the time when a positive BOLD signal reached its maximum and the time to baseline as the time when the BOLD signal returned to the baseline, both measured referring to the onset of stimulation.

Here, a multiple regression analysis [23] was also conducted to investigate the variance of the brain dynamic in response to the stimulus of different current intensities and distinct modalities (i.e., IFC vs. ACS). This group-level comparison was implemented in the SPM12 software by inputting beta-maps of respective subjects computed from the first-level analysis and designating stimulation intensity as the regressor.

Statistical processing was performed using MATLAB 2022 and Python 3.8. Family-wise error correction was conducted for the voxel-based analysis of BOLD-fMRI. Bonferroni corrections or false discovery rate corrections were applied to correct *p*-values for multiple comparisons. Values are shown as the mean ± standard error of the mean or as scatter diagrams with crossbars showing the average values.

## 3. Results

We initially confirmed the position of the implantation by examining the T2-weighted structural and EPI image taken from the preliminary experiment. As demonstrated by Figure 1B, the stimulation electrode was located at the targeted M1 region as expected, with limited tissue damage. Artifacts induced by the electrode could be restricted exclusively in the region surrounding the wire, demonstrating the capability of a concurrent stimulation experiment with fMRI image taking. Animals with imprecise implantation of the electrode (e.g., deep to the striatum, *n* = 3) were excluded from the data analysis. In addition, data with a significant motion artifact (which could not be removed by our motion correction method, *n* = 2) were not included.

### 3.1. Spatial Contribution of Whole-Brain BOLD Activation Evoked by Stimulation

Then the whole-brain responses were investigated in response to both ACS and IFC stimulation based on BOLD-fMRI. We eventually accumulated data from *n* = 8 rats with a random choice of the priority for undergoing ACS and IFC stimulation, with an interval of 15 min for recovering between two stimulation groups.

Based on the GLM statistical analysis, we identified the brain areas significantly modulated by the electrical stimulation with voxel resolution. The activation maps of each session were depicted, indicating the brainwide BOLD activity changes evoked by respective stimulation. Here, we presented a representative result, where evident activation signals were observed in the last two sessions but not in the first session undergoing ACS (Figure 2, Top), suggesting that we successfully matched stimulation strength for recruiting neural responses. Activation maps demonstrated that activation (positive BOLD variations) signals were primarily distributed in the superficial cortical regions, which comprised M1 on the bilateral side of the stimulation, the bilateral secondary motor cortex (M2), the ipsilateral primary somatosensory cortex (S1), and the secondary somatosensory cortex (S2). We also observed a positive BOLD response in the striatum (CPu) in both stimulation modalities. Following the increment of the current intensity, we observed a broader distribution of the activated voxels, including BOLD activity in contralateral S1 and bilateral retrosplenial cortex regions in some subjects (*n* = 2). Note that voxels of S1 that exhibited positive BOLD activities covered extensive structures comprising the barrel field, forelimb region, and hindlimb region. In addition, we observed negative BOLD signals in the hippocampal area of two rats (*n* = 2) corresponding to the subiculum regions.

However, IFC stimulation could evoke significant BOLD activity exclusively when an interferential wave was applied at an intensity surpassing 0.6 mA. No evident BOLD signal changes were observed when the stimulus was employed with a comparable current intensity with the ACS group (lower than 0.6 mA), and activation signals were shown to occur merely in restricted regions in bilateral M1 when the current intensity rose to 0.8 mA (Figure 2, Bottom). Notably, using a high-intensity stimulation, the overall distribution of the activation signals elicited by IFC followed that induced by ACS, where signals propagated horizontally from the stimulated area to the contralateral structures (M1, M2). These results are consistent across all the rats, where a larger number of voxels exhibited increased BOLD activity when ACS was applied in contrast with that elicited by IFC stimulation (for ACS, an average of 1096 voxels with 0.5 mA and 723 voxels with 0.4 mA; concerning IFC, an average of 428 voxels with 0.8 mA and 205 voxels with 0.6 mA).

It is noteworthy that the vanished activity in the regions surrounding the electrodes could be explained by the deterioration of the signal owing to the electrode-induced artifacts, as a result of which the BOLD response in the target region (i.e., ipsilateral M1 area) could not be confirmed for a subset of subjects (*n* = 3). Nevertheless, the increment of the signal intensity (paired *t*-test, compared with baseline signal, p<0.001) was identified in the area encompassing the electrodes for the remaining subset of animals (*n* = 4, Figure 3), indicating that the applied invasive modality was able to directly recruit neural activation in the target region and consequently modulating functionally connected brain regions through the neural network.

### 3.2. Characterization of BOLD Time Courses

We further investigated the BOLD signal changes (percentage) against scan time in response to different stimulation modalities. Regions of interest (ROIs) of ACS stimulation were selected as bilateral S1 and contralateral M1, and the time series was extracted based on the GLM statistic activation map, averaged across all the voxels of respective functional regions. The results of ACS stimulation sessions with intensities of 0.3, 0.4, and 0.5 mA are presented here. As demonstrated by Figure 4, an identical BOLD signal increment was identified in target regions and was synchronized with the stimulation period, suggesting the production of a consistent response to the stimulation. By comparing the peak intensities across all stimulation blocks, a stronger BOLD activity was confirmed in accompany with the increase of the stimulation current, as presumed (one-way repeated measures analysis of variance (ANOVA) tests with Tukey post hoc analysis, p<0.01).

The analogous tendency was confirmed during IFC stimulation sessions, where the intensity of the evoked positive BOLD response increased monotonically with the current intensity gradient. The BOLD profile was comparable with the ACS modality; however, the requisite stimulus intensity to elicit a prominent BOLD response differentiated between the two modalities. Moreover, the overall magnitude of BOLD responses to IFC stimulation was smaller even when a relatively strong electric current was applied.

The variance of response amplitude was confirmed by an intermodality comparison of the peak intensities of the BOLD time course (Figure 5, left panel, one-way repeated measures ANOVA tests with Tukey post hoc analysis, p<0.05); at a comparable stimulus intensity (e.g., 0.4 vs. 0.4 mA, 0.5 vs. 0.6 mA), IFC stimulation was unlikely to elicit an analogous BOLD response relative to ACS, implying that IFC stimulation might not be as efficient as conventional brain modulation methods. In addition, we inspected temporal profiles of the induced BOLD response where no latency was observed following the onset of the stimulus, indicating that all these regions were activated almost simultaneously through inter- and intrahemispheric cortical pathways. The IFC stimulation could elicit a more rapidly ascending BOLD activation in contrast with ACS, which could be corroborated by comparing the averaged times to peak of the BOLD time series (Figure 5, middle panel, significant change in bilateral S1 and M1; ANOVA tests with Tukey post hoc analysis, *: p<0.05, **: p<0.01). Conversely, a BOLD response triggered by IFC stimulation recovered to the resting state more slowly than ACS by examining the time to baseline (from the onset of the stimulus) across sessions (Figure 5, right panel, significant change in bilateral S1 but not M1; ANOVA tests with Tukey post hoc analysis, *: p<0.05, **: p<0.01).

### 3.3. Stimulation Effects Modulated by Stimulus Intensity

Multiple regression analysis was conducted with stimulation intensity selected as a predictor, and t-contrasts were utilized to test the individual effects for significance. As shown by Figure 6, the statistical map of the brain responses to ACS and IFC stimulation indicated that both modalities were moderated by the intensity of the stimulation current. We observed that the neural activity induced by ACS was positively correlated with the stimulus amplitude in the bilateral S1 and ipsilateral CPu regions. This amplitude-dependent response was observed in ipsilateral S1 and contralateral M1 with respect to IFC stimulation but not found in CPu.

In summary, we here conducted a brainwide BOLD response to both invasive IFC stimulation and ACS. We found that the activation threshold of IFC (0.4 mA) was higher than that of ACS (0.3 mA), and to induce an analogous BOLD intensity, at least a twofold larger stimulus current (0.4 mA for ACS vs. 0.8 mA for IFC) was requisite. In addition, the distribution of voxels with activation signals was more restricted in IFC sessions, and distinct BOLD temporal characteristics were confirmed between two modalities.

## 4. Discussion

Grossman et al. proposed TI as a potential noninvasive brain modulation approach to recruit neural activities in the subcortical brain regions [1]. The premise of the TI stated that when multiple high-frequency currents are superposed to the brain, an interferential current will be generated because of amplitude modulation, and the envelope (which should be oscillating at the different frequency of the two source currents and is also known as “beat”) can be exploited for neural modulation [1]. However, notwithstanding the promising concept, the transition from theory to clinical application appears to be challenging [3,24], and the stimulation effect of TI was proved to be confounding [11,25]. Therefore, in this study, we aimed to address the fundamental issues of this method: whether interferential current (the basis of TI) could elicit brain responses just as done by conventional brain stimulation modalities. If so, what are the acute brain dynamics relative to this stimulus? Here, a straightforward paradigm was employed where functional mappings were evaluated upon an invasive stimulation using both interferential electric (IFC) and alternating sinusoidal current (ACS). In TI stimulation, by aligning the electrodes, the depth of the envelope will reach its maximum exclusively in the target regions and consequently focally modulate neural activities; therefore, we conclude that, nevertheless, the spatial electric field distribution generated by an invasive modality did not completely reproduce that produced by transcutaneous IFC (i.e., TI stimulation). It should be analogous exclusively in the target region. Consequently, our results could provide pieces of evidence for the application of TI as a novel brain stimulation modality.

It is noteworthy that the concept of leveraging interferential current for stimulation was proposed decades earlier and is more widely known as IFC therapy (ICT) [26]. However, this method was generally exploited for treating musculoskeletal pain [27] rather than modulating cerebral functions. Therefore, we initially examined whether IFC can evoke suprathreshold neural activities inside the brain, possibly indicated by the regional hemodynamic response taken from fMRI scans. An interferential current was directly delivered to M1 to replicate our previous motor-cortical stimulation paradigm [16], by which we anticipated that brain regions in the cortico-striato-thalamo-cortical circuit would be activated in response to the stimuli as demonstrated by studies using conventional stimulation modalities [28,29]. Indeed, positive BOLD signal changes were observed in cortical and subcortical regions on bilateral aspects of the brain, indicating signal propagation through both interhemispheric and intrahemispheric pathways, which suggested that IFC stimulation is capable of inducing regional neural responses and modulating brain networks consequently. Furthermore, the observed response is proportionally increased and correlated with stimulus intensity (multiple regression analysis).

However, we also observed that the threshold for significantly activating a neural response using IFC was higher relative to stimulation through a low-frequency alternating current. Another observation demonstrated by the activation maps was that the activation signals induced by IFC were restricted even when a relatively strong electric current (0.8 mA) was applied. Moreover, the requisite stimulus intensity for invasive IFC to the induced comparable BOLD response intensity against ACS is at least twofold larger (0.4 mA for ACS vs. 0.8 mA for IFC). These results imply that suprathreshold stimulation using IFC is not as efficient as using a conventional form of stimulus. Therefore, we suggest that a more attentive discussion should be conducted when considering the application of TI. This is because in the context of TI stimulation, subcortical brain areas were decided as the target for the direct recruitment of neural activity as in the deep brain stimulation (DBS) modality; however, the electric field intensity induced by transcranial modalities can be much weaker than invasive approaches due to the rapid attenuation of the induced field [30]. Here, the comparison was conducted between IFC and ACS, but the results can be analogous to the contrast of TI versus tACS. Nevertheless, a conventional tACS modality could possibly manipulate a BOLD signal; this effect can be too weak to be detected by fMRI [31]. Considering the higher requisite activation threshold of IFC, a much higher stimulus current might be required for TI to induce a regional BOLD response.

The same inference was discussed in the computational comparison between tACS and TI but not in the context of an in vivo experiment. For a conventional tACS modality, the commonly applied current intensity is about 2 mA, which could produce an electric field of 0.8 V/m in the human brain scale [32]. However, based on computational simulation, previous results suggested that the requisite temporal interferential electric field for an acute neural response in a mouse ranges from 60 to 350 V/m [3,15], equivalent to an applied stimulus current exceeding 160 mA above the scalp [33]. Therefore, the modulation of deep brain regions using a temporal interferential field could be challenging, and our results, which inspected a hemodynamic response, provided a corroboration to this inference. We showed that invasive IFC stimulation is unlikely to reproduce the modulation effect of invasive ACS at an analogous stimulus intensity, which brings implications when the modality is switched from an invasive solution to a noninvasive strategy [34], especially considering that the primary target of TI stimulation is subcortical structures of the brain.

Additionally, the temporal characteristics of the BOLD time course of the respective stimulation sessions suggested that IFC stimulation could elicit a faster (baseline to peak time) and prolonged (time of recovering to baseline) hemodynamic response. The differentiated BOLD signal could be explained by activating distinct cell types in the brain. Studies have demonstrated that a BOLD response could be constituted by hemodynamic information from not only local excitatory neurons but also inhibitory neurons, astrocytes, and vascular cells [35,36]. We believe that IFC could induce the recruitment of inhibition neurons, the hemodynamic response function of which has been demonstrated to reach the peak earlier than that of excitation neurons [37]. Consequently, this inhibition (which is more likely to occur in response to stronger stimuli [38]) could potentially reduce the magnitude of the positive BOLD elicited by the stimulus. Note that a pivotal premise of TI stimulation is that neural membranes do not respond to high-frequency stimulation, or more precisely, they are more sensitive to a slow oscillating field [1]. However, Kilohertz-frequency stimulation was also demonstrated to selectively activate inhibitory interneurons [39,40], which should account for the inhibition effects using IFC stimulation, considering the fundamental high-frequency component of the current wave. In respect of a prolonged BOLD response, however, the possible interpretation would be the occurrence of afterdischarge [17] or the increased vascular and/or astrocytic activity, which resulted in the decoupling between neuronal and hemodynamic signals [41]. Note that notwithstanding the prominent spatial resolution provided by fMRI, some results in this paper are still indirect due to the limited temporal resolution of this approach and, therefore, should be interpreted more carefully. A further consolidation of the results can be conducted by exploiting electrophysiological recordings of the neurons exclusively in the target region simultaneously with the stimuli.

TI stimulation was initially proposed as a potential noninvasive DBS modality; therefore, suprathreshold stimulation effects are expected in clinical application. However, theoretical and experimental evidence suggested that noninvasive transcranial electrical stimulation approaches are unlikely to affect neuronal circuits directly and instantaneously [3,13], which implicated the application of TI in clinical neurostimulation therapies [25]. However, studies have also highlighted the possibility of transcranial electrical stimulation to provide subthreshold neuromodulation effects rather than a direct activation of neurons [14,42]. Therefore, a future application of TI stimulation should take these aspects into account and perhaps consider other possible neuromodulatory effects of noninvasive stimulation. Another potential application of TI could be motor cortical stimulation that targets a more superficial tissue of the brain [8,43]. Because alternating current has been demonstrated to have the capability of regulating neural response [44,45], an analogous effect might be anticipated, exploiting interferential current transcranially [46] or even an interferential magnetic field [47].

*Limitation*: In this study, we did not investigate the acute neural response subject to a high-frequency alternating current solely, in contrast with interferential current. However, potential side effects engendered by a high-frequency oscillating field should be cautiously discussed, such as the phenomenon of conduction block [48] or desynchronization of neural firing [49], because the superficial cortex distant from the target will be exposed to the high-frequency oscillating field in the transcranial TI scenario. Another concern is the implantation of the electrode, which could possibly cause an adverse incident, such as intracranial hemorrhage or infections [50,51], especially in the chronic DBS or motor cortex stimulation protocols. However, we argue that in our acute experiments, the intruding electrode did not induce severe blooding or aberration of the brain tissue (Figure 1), and the robust brain network response induced by this paradigm (which has been validated by precedent studies [16,17,52]), cannot be simply explained by a hyperthermia effect.

## 5. Conclusions

In summary, here, we investigated acute brain responses to invasive IFC stimulation compared with the analogous modality using ACS. We conclude that IFC stimulation is capable of inducing an intensity-dependent suprathreshold activation of the brain tissue. However, a higher activation threshold indicates that this approach might not be as efficient as conventional brain modulation methods, especially when considering TI stimulation as an alternative to the transcranial DBS approach. Furthermore, distinct patterns of the brain response against IFC indicate a possible intermingled recruitment of different types of cells, which can be the main focus of future study. We believe that this study will help us to understand some fundamental mechanisms and characterization underlying this novel technique, consequently facilitating the translation of theoretical studies to clinical therapy.

## Figures and Tables

**Figure 1 brainsci-13-01317-f001:**
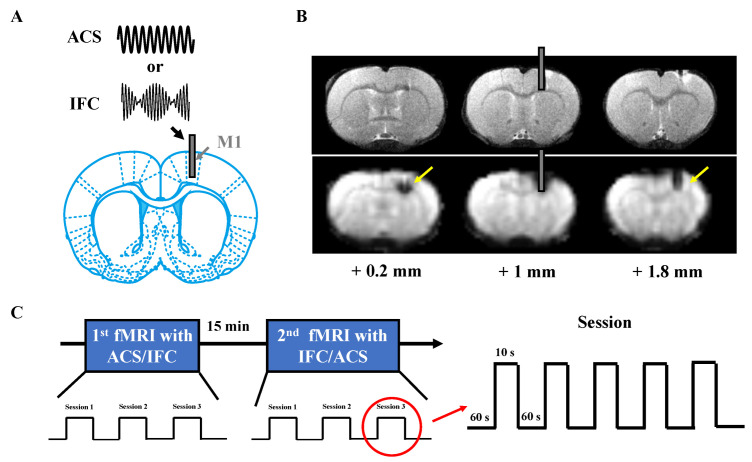
Experimental protocol for conducting concurrent stimuli fMRI experiment. (**A**) A homemade electrode was implanted into the M1 area of the rat brain, through which a stimulus was directly exerted on the cortical neurons. The stimulus was composed of either a 10 Hz sinusoidal current wave (as for ACS) or an IFC wave with a base frequency of 2 kHz and a modulated frequency of 10 Hz (IFC). (**B**) Representative T2-weighted anatomical and EPI images presenting brain slices from posterior 0.2 mm to anterior 1.8 mm. The rectangle in gray color represents the embedded stimulation electrode. The yellow arrow points to an artifact induced by the presence of the electrode. (**C**) Diagram showing the timing of the BOLD data acquisition: In each session, a stimulus was applied as a block design with a total duration of 6 min and 50 s. All the animals underwent both stimulation protocols with a random order.

**Figure 2 brainsci-13-01317-f002:**
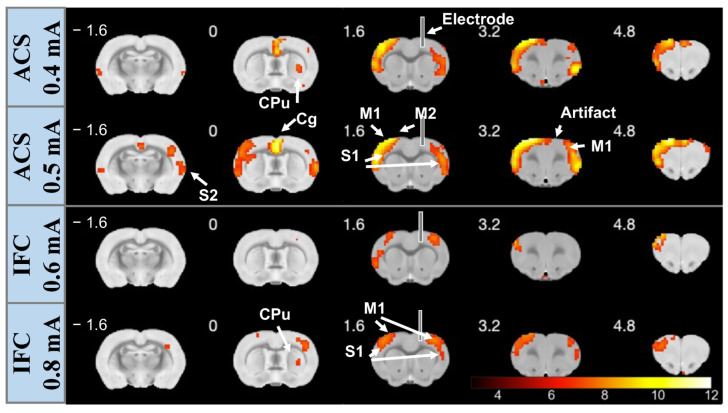
A representative GLD-based fMRI analysis in response to a respective stimulus with an increment of current intensity. Activation maps demonstrate brain slices positioned from −1.6 to 4.8 mm anteroposterior with current intensity: 0.4 mA (**Upper-Top**) and 0.5 mA (**Upper-Bottom**) for ACS, and 0.6 mA (**Lower-Top**) and 0.6 mA (**Lower-Bottom**) for IFC. The label of the respective slice indicated the slice position related to the bregma (mm), and the color bar indicated the statistic T-value from GLM analysis with a restriction of p<0.001. M1: primary motor cortex; S1: primary somatosensory; M2: secondary motor cortex; S2: secondary somatosensory cortex; CPu: striatum.

**Figure 3 brainsci-13-01317-f003:**
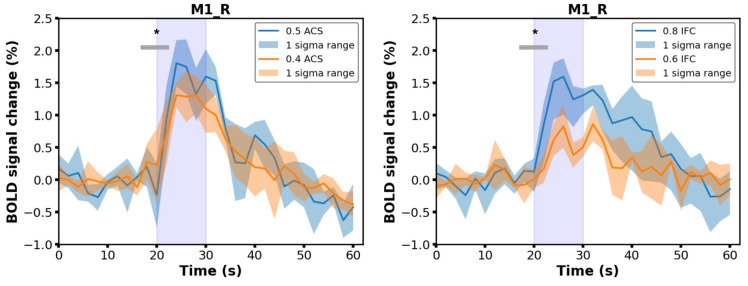
Time course of the BOLD variation in the target area (M1), averaged across a subset of subjects (*n* = 4) where the activation signal in the region encompassing the electrode could be identified based on the GLM analysis. The time course was averaged across stimulation sessions, and the signal during the stimulus period was shown to be significant to the baseline when the stimulus intensity surpassed 0.4 mA using ACS or 0.6 mA using IFC stimulation (Purple color: stimulus on period; * paired *t*-test, p<0.001).

**Figure 4 brainsci-13-01317-f004:**
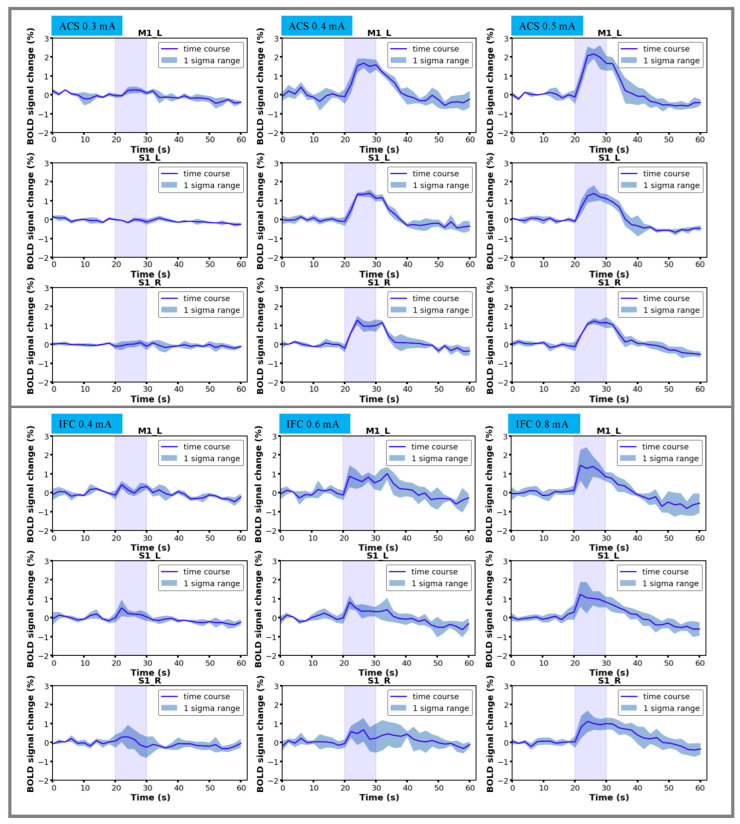
Mean time series of all stimulation protocols extracted from selected regions of interest based on GLM-BOLD activation T-value maps. The time series was initially averaged across stimulation sessions and succeedingly averaged across all the animal subjects. (**Top**: BOLD time course under ACS; **Bottom**: BOLD time course under IFC stimulation. M1_L: contralateral primary motor cortex; S1_L: contralateral somatosensory cortex; S1_R: ipsilateral somatosensory cortex; Purple color: stimulus on period).

**Figure 5 brainsci-13-01317-f005:**
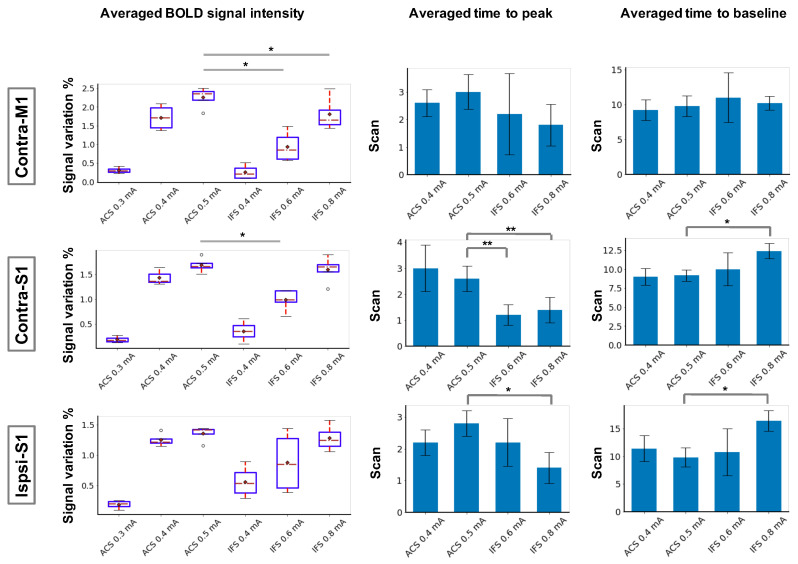
BOLD signal characteristics of contralateral M1 and bilateral S1 in response to IFC stimulation and ACS. (**Left column**) averaged peak intensity of the BOLD response. *: p<0.05, one-way repeated measures ANOVA tests with Tukey post hoc analysis. (**Middle column**) averaged time to peak of the BOLD response. Significant difference in bilateral S1 and M1; *: p<0.05, **: p<0.01. **Right column**: averaged time for returning to baseline. Significant difference in bilateral S1 and M1; *: p<0.05.

**Figure 6 brainsci-13-01317-f006:**
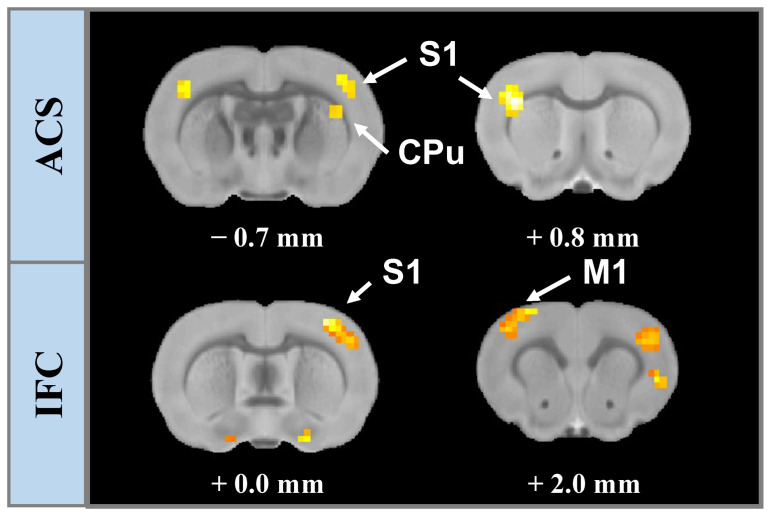
Results of the second-level multiple regression analyses. The statistical maps of the brain responses induced by both IFC (**lower panel**) stimulation and ACS (**upper panel**) were demonstrated to be positively associated with stimulus intensity. The label of the respective slice indicated the slice position related to the bregma (mm). M1: primary motor cortex; S1: primary somatosensory; CPu: striatum.

## Data Availability

The datasets (GENERATED/ANALYZED) for this study can be accessed by contacting the leading authors.

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
