# Peer review of "Brain Response to Interferential Current Compared with Alternating Current Stimulation"

_brainsci, 2023, doi:10.3390/brainsci13091317_

Round 1
Reviewer 1 Report
Comments and Suggestions for Authors
In this study, the authors have tried to find a way to support pieces of evidence on the effectiveness of Interferential Current Stimulation (ICS) on the brain. But the study design is not appropriate. The applied method is completely invasive, and could not be classified as an Interferential Current Stimulation (ICS) but both applied stimulations are a form of deep brain stimulation.
The authors could report this work as the effects of applying different electrical pulse waveforms and parameters (frequency and ...) and their effects.
Author Response
We appreciate your comment, please see the attachment for the response and revision.

Reviewer 2 Report
Comments and Suggestions for Authors
What criteria were used to determine the sample size?
The purpose of the study in lines 133-138 must precede the materials and methods
The design of the study is not clear. It is necessary to clarify how many experimental animals underwent ACS and IFS.
In the methods and figures, interference current stimulation was carried out only in one area. But interference stimulation is characterized by simultaneous electrical stimulation in many areas. Please explain.
The materials should describe in detail the methods for studying the response of the contralateral primary motor cortex, the response of the contralateral somatosensory cortex, and the response of the ipsilateral somatosensory cortex.
Figure 7. Incomprehensible. value must be specified ACS, IFC, CPu, T1, S1. What do the numbers (0.7, 0.8, 0.0, 0.2) mean?.Where can we find the intensity of the stimulus in this figure?
Author Response

(The authors gave the same response as above.)

Reviewer 3 Report
Comments and Suggestions for Authors
ABSTRACT
- it would be more beneficial if the authors would add more information in the abstract
- the conclusion part of the abstract lacks important information
INTRODUCTION
- what is the meaning of "capacitive properties"?
- I feel that the sentences are too long and contain a lot of valuable information that could be forsaken easily because of the ambiguity and length of the sentences
- also, you should briefly mention the main aim of the study and the conclusions that you want to obtain
RESULTS
- in this section, i would like to see a last paragraph that will contain some ideas from all the experiments.
DISCUSSIONS
- this chapter is missing the limitations of the study
- also, i would want in this section at least a table that will contain differences and sections in common between your study and other studies in order to enhance the meaning of the written information
- furthermore, talk about extending this type of experiment and the meaning of this study on the discoveries of our century.
CONCLUSIONS
-this section is missing information, only 6 lines are not enough for this kind of study
REFERENCES
-more references would be needed in order to sustain your manuscript
Author Response

(The authors gave the same response as above.)

Round 2
Reviewer 2 Report
Comments and Suggestions for Authors
1. No doubt the manuscript has been improved. However, some issues have not been fully resolved.
2. The most important issue is the determination of the sample size. Authors should look for similar work in previous studies and determine expected outcomes, and use special formulas to determine the smallest number of patients in each group. more ever, The text should indicate:
· the primary endpoint (measurement) that allows the calculation of the minimum expected difference.
· Power value = %.
· Expected significance level (p-value) .
Authors can find a lot of information on these sites.
https://clincalc.com/stats/samplesize.aspx
and
https://www.sealedenvelope.com/power/binary-superiority/
3. In the design, authors should inform readers about the total number of experimental animals. It is also necessary to clarify the number and reasons for exclusion. After treatment, it is important to provide information about animals withdrawn from the study due to intolerance or side effects. Since the experimental animals underwent the same courses with the use of two methods of treatment, it is necessary to indicate the timing of the application of each method of treatment. It is desirable to use pictures and drawings.
Reviewer 3 Report
Comments and Suggestions for Authors
Congratulations!
Author Response
We appreciate your acceptance. We have modified our manuscript based on some other comments, please confirm the newly uploaded manuscript.